# Rate and timing of cortical responses driven by separate sensory channels

Hannes P Saal[1]*, Michael A Harvey[1,2], Sliman J Bensmaia[1,3]*

[1]Department of Organismal Biology and Anatomy, University of Chicago, Chicago, United States; [2]Department of Neurological Surgery, Vanderbilt University Medical Center, Nashville, United States; [3]Committee on Computational Neuroscience, University of Chicago, Chicago, United States

**Abstract** The sense of touch comprises multiple sensory channels that each conveys characteristic signals during interactions with objects. These neural signals must then be integrated in such a way that behaviorally relevant information about the objects is preserved. To understand the process of integration, we implement a simple computational model that describes how the responses of neurons in somatosensory cortex—recorded from awake, behaving monkeys—are shaped by the peripheral input, reconstructed using simulations of neuronal populations that reproduce natural spiking responses in the nerve with millisecond precision. First, we find that the strength of cortical responses is driven by one population of nerve fibers (rapidly adapting) whereas the timing of cortical responses is shaped by the other (Pacinian). Second, we show that input from these sensory channels is integrated in an optimal fashion that exploits the disparate response behaviors of different fiber types.

*For correspondence: hsaal@ uchicago.edu (HPS); sliman. bensmaia@gmail.com (SJB)

**Competing interests:** The authors declare that no competing interests exist.

## Introduction

Perception reflects the seamless integration of signals from a variety of sensory receptors that respond preferentially to different aspects of the environment. The classic example of such integration is color vision, where input from different cone receptors is integrated to extract a specific feature of objects, approximately corresponding to their absorption spectrum (*Gegenfurtner, 2003*). Signals are also combined across different sensory modalities, and this integration process is often optimized to extract stimulus information (*Ernst and Banks, 2002*). Given that different sensory channels encode different stimulus features and exhibit different response properties, the central nervous system is faced with the challenge of how to integrate such disparate input signals.

The sense of touch is mediated by at least three main classes of mechanoreceptive afferents, each of which responds to different aspects of skin deformation (*Johnson, 2001*): slowly-adapting type I (SA1) afferents are most sensitive at low frequencies (<10 Hz), rapidly adapting (RA) afferents at intermediate frequencies (in the so-called flutter range, from 10 to 50 Hz), and Pacinian (PC) afferents at high-frequencies (peaking in sensitivity at around 250 Hz). However, the frequency sensitivities of the different afferent classes overlap considerably (*Figure 1A*), and natural stimuli typically engage more than one class (*Saal and Bensmaia, 2014*). Not surprisingly, then, information about most tactile events is carried by signals from several afferent classes (*Saal and Bensmaia, 2014*; *Johansson and Flanagan, 2009*; *Abraira and Ginty, 2013*), and this input is integrated across submodalities even at the first stage of cortical processing, namely primary somatosensory cortex (S1) (*Pei et al., 2009*; *Carter et al., 2014*). Importantly, S1 neurons convey information about contacted objects not only in the strength of their responses but also in their timing (*Harvey et al., 2013*; *Zuo et al., 2015*). For example, the amplitude of skin oscillations—such as those elicited during the exploration of textured surfaces—is encoded in the strength of the cortical response whereas

**eLife digest** Our sense of touch depends upon receptors in our skin that send signals to the brain about the objects with which we interact. Different types of touch receptors respond in different ways when we grasp and manipulate objects; for example, by altering the strength of their response or its timing.

Saal et al. have now investigated how neurons in a part of the brain called the somatosensory cortex, which processes touch signals from the hand, respond to the information from the different receptor types. First, recordings were made of the electrical activity of the touch receptors and the neurons in the brain of monkeys. Using this data, Saal et al. built computer models that allow the response of neurons in the brain to be predicted from the responses of the touch receptors.

The models showed that signals from different types of touch receptors shape the response of neurons in the brain in different ways. One receptor type controls how strong a neuron's response will be, while another one controls the precise timing of the response. Further investigation revealed that this way of combining the signals from the different receptors preserves as much information as possible about objects and thereby helps the brain to process information acquired by touch quickly and efficiently.

Future experiments will examine how touch is represented in two structures that lie between the receptors and the somatosensory cortex: one in the brainstem, the other in a brain region called the thalamus.

their frequency is encoded in its timing. To achieve such multiplexing requires that afferent signals be integrated in a principled way.

Here, we measure the responses of S1 neurons to a wide variety of simple and complex vibrotactile stimuli and reconstruct the responses of afferent populations to these same stimuli. We then implement a simple model that allows us to examine how responses of the different afferent populations combine to drive responses in cortex. First, we find that most S1 neurons integrate information from multiple afferent classes. Second, signals from the different classes drive cortical responses in very different ways: RA afferent input exerts an excitatory influence on S1 responses and is the primary determinant of their firing rates; in contrast, PC input exerts a balanced excitatory and inhibitory influence which sharpens the timing of cortical responses without having a major impact on the rates. Finally, we show that the process of integration maximizes information transmission across the range of stimulation conditions that might be encountered during everyday interactions with objects.

## Results

We recorded the responses of 18 peripheral tactile afferents (14 RA and 4 PC) and 118 cortical neurons in areas 3b (49) and 1 (69) of S1 to a variety of vibrotactile stimuli, ranging from simple sinusoids to complex bandpass noise with frequencies between 50 and 800 Hz and amplitudes ranging from sub-micrometer to more than 500 µm. Stimulation conditions spanned those that might be experienced when we scan a textured surface (*Bensmaïa and Hollins, 2003*; *Bensmaïa and Hollins, 2005*; *Weber et al., 2013*; *Manfredi et al., 2014*). As these stimuli drove RA and PC to the exclusion of SA1 afferents (*Muniak et al., 2007*), we focused the analysis on these two afferent populations. The goal was to quantitatively predict S1 responses based on afferent responses. Because the stimuli used in the peripheral experiments were analogous but not identical to those used in the cortical experiments, we used simple but precise models of afferent responses (derived from recorded responses, *Figure 1—figure supplement 1*) to reconstruct RA and PC population responses to the stimuli used in the cortical experiments (*Figure 1C,D*). To examine how input from these different populations is integrated in cortex, we describe the transformation from the periphery to cortex as a linear filter over the RA and PC population responses whose output is subsequently rectified (based on the linear-nonlinear-Poisson framework (LNP), *Figure 1B*). This approach allows us to examine the filters estimated for each cortical neuron to assess the contribution (if any) of each afferent population to that neuron's response and to characterize the dynamics of this integration. To the extent

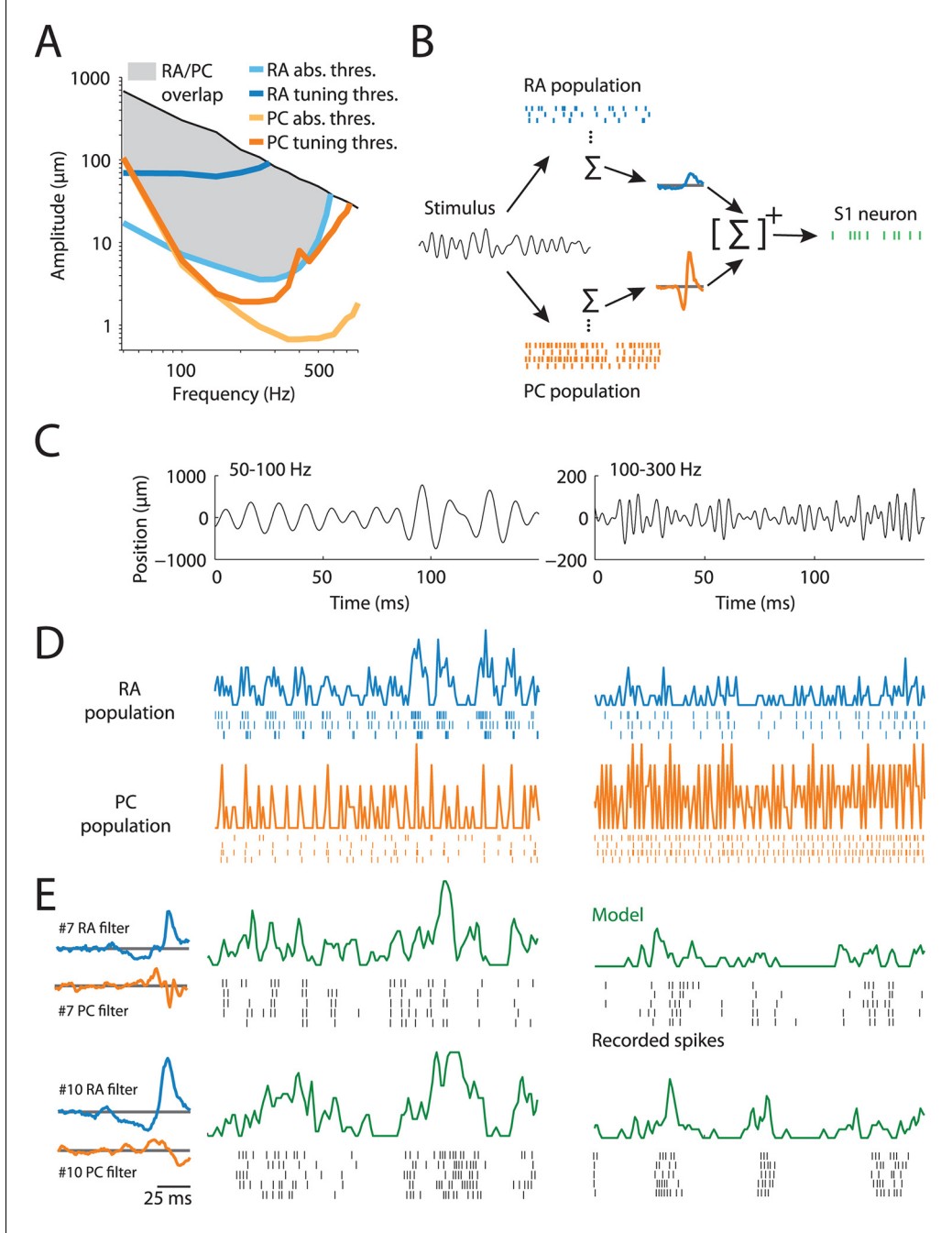

**Figure 1.** Characterizing the transformation between peripheral and cortical responses. (**A**) Frequency–amplitude pairs that elicit responses in rapidly adapting (RA) and Pacinian (PC) afferents. Light blue and orange lines indicate absolute response thresholds for RA and PC afferents (as determined by the peripheral model), respectively, while the dark blue and orange lines denote tuning thresholds (i.e., eliciting one spike on each stimulus cycle). The black line corresponds to the highest amplitude the stimulator could deliver at a given frequency and likely exceeds the maximum amplitude that one is liable to encounter during every day tactile experience. While RA and PC afferents are maximally sensitive at different frequencies, most frequency–amplitude pairs elicit responses from both afferent classes (indicated by the shaded region). (**B**) Illustration of the model that describes the transformation from peripheral to cortical responses. A broadband noise stimulus activates both RA and PC afferents. RA and PC population responses are each convolved with a temporal filter, then summed and rectified to culminate in the responses of individual S1 neurons. (**C**) Two examples of vibratory stimuli. (**D**) Simulated RA (blue) and PC (orange) population firing rates along with spike trains of a subset of neurons in the population. See *Figure 1—figure supplement 1* for details on how the peripheral population models were validated. (**E**) Recorded spikes (black

*Figure 1 continued on next page*

*Figure 1 continued*

ticks) over five stimulus repeats and model predictions (green traces) for two cortical neurons, whose RA and PC filters (plotting filter magnitude over time) are shown on the left. Cortical neurons differ in their response properties, such as burstiness and temporal precision, and the model captures these broad differences with different RA and PC filters. See *Figure 1—figure supplement 2* for assessment of the general predictive power of our model.

The following figure supplements are available for figure 1:

**Figure supplement 1.** Simulation of populations of RA and PC afferents.

**Figure supplement 2.** Prediction accuracy of fitted LNP model.

that this simple model captures different response properties of cortical neurons, the respective filters for these neurons will be different (*Figure 1E*).

To test the model's performance, we compared predicted and recorded cortical responses at different timescales. We performed this analysis on a separate data set (consisting of sinusoidal stimuli), to estimate how well the model would generalize to stimuli that differed considerably from the noise traces used to fit the model. We found that the model captures the cortical responses well on both coarse and fine timescales as assessed by calculating the correlation coefficients between the predicted and recorded responses. Indeed, across neurons, the model captured 67% ( ± 2% standard error of the mean) of the predictable variance (*Ramirez et al., 2014*) for 1 s bins, and 35% ( ± 2%) for 5 ms bins. In other words, the model was able to predict neuronal responses with considerable accuracy given their inherent variability.

## RA and PC input is integrated differently

We found that most cortical neurons in areas 3b and 1 receive input from both RA and PC afferents, as evidenced by the significant contribution of both to the overall prediction (*Figure 2A*).

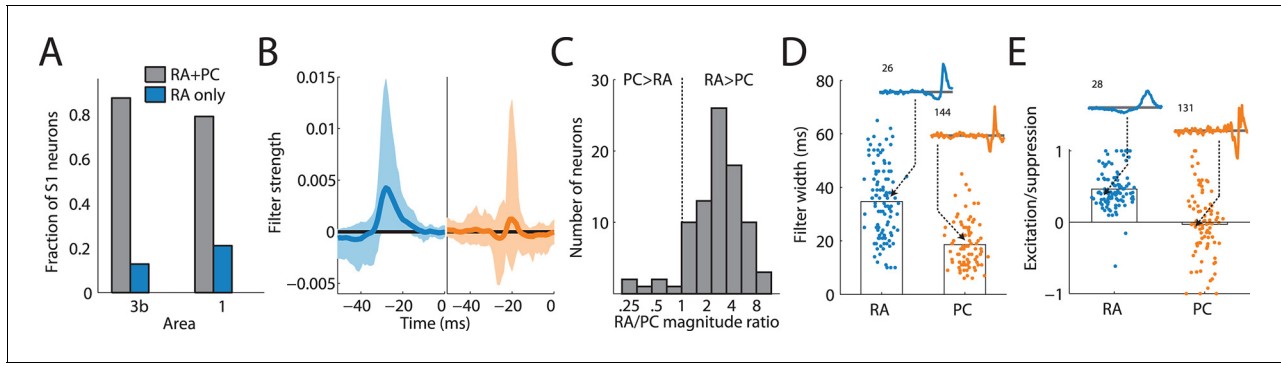

**Figure 2.** Rapidly adapting (RA) and Pacinian (PC) input is integrated differently. (**A**) Fraction of neurons in areas 3b and 1 that receive input from both RA and PC populations (grey bar) or from only the RA population (blue) as assessed by whether input from the respective class is weighted more highly than might be expected by chance. Most S1 neurons receive input from both RA and PC afferents. (**B**) Mean RA and PC filters (solid lines) and the 5th and 95th percentile of filter values across all S1 neurons indicated by the light-shaded regions. RA and PC filters are systematically different. (**C**) Ratios of RA and PC filter magnitudes for neurons integrating information from both afferent classes. RA input tends to be weighted more strongly. (**D**) Width of RA and PC filters. RA input tends to be integrated over longer timescales than does PC input. Insets show example filters that are close to the population average (over 100 ms). (**E**) Excitation indices for all RA and PC filters, where 1 denotes purely excitatory input, 0 balanced input, and -1 purely suppressive input. RA filters almost always lead to a net positive excitation of the cortical neurons, while the PC filters are more diverse, on average balanced, and often suppress cortical response. Insets show example filters that are close to the population average (over 100 ms). See *Figure 2—figure supplement 1* for validation of the analysis.

The following figure supplement is available for figure 2:

**Figure supplement 1.** Accuracy of linear filter estimation.

Furthermore, the linear filters obtained for each afferent population, while similar across cortical neurons, were consistently different from each other (*Figure 2B*): First, while the magnitude of the PC filters was correlated with that of the RA filters ($r = 0.41$), PC filters were consistently of lower magnitude (*Figure 2C*, $t(70) = 6.4$, $p < 0.01$, paired t-test), with the RA contribution on average about three times stronger than the PC one. Second, RA filters were much more extended in time (with a mean above 30 ms) than were PC filters (with a mean below 20 ms) (*Figure 2D*, $t(188) = 9.6$, $p < 0.01$, two-sample t-test). Third, while the net contribution of the RA input to the cortical response was positive, PC input was on average balanced; that is, excitatory and suppressive components of the PC filters were of equal magnitude. In fact, for about half of cortical neurons, the PC contribution was net suppressive (*Figure 2E*); the difference in the net contribution of RA and PC input was significant ($t(188) = 16.1$, $p < 0.01$, two-sample t-test). PC filters were also much more diverse and spanned the range of this metric (from -1 to +1), in contrast to RA filters, which, as mentioned above, tended to be net excitatory ($>0$).

## RA input drives cortical firing rates

The different filter shapes for RA and PC input reflect profound differences in the way the two afferent populations shape cortical responses. Indeed, the firing rates of individual S1 neurons are almost entirely determined by RA input as evidenced by the fact that a model based on RA input alone can predict S1 firing rates—averaged over 1 s long periods—almost as well as RA and PC responses combined; this was true for responses to both noise and sinusoids (*Figure 3*). PC input alone, on the other hand, cannot account for the observed cortical firing rates. Indeed, these would be much more frequency-dependent if driven by this population of nerve fibers.

## PC input determines precise spike timing

We have previously shown that the responses of mechanoreceptive afferents are highly precise and repeatable, and their timing conveys information about the frequency composition of skin vibrations (*Mackevicius et al., 2012*) and about the identity of textures scanned across the skin (*Weber et al., 2013*) at resolutions on the order of milliseconds. Similarly, the temporal patterning of responses in somatosensory cortex is precise and informative down to a millisecond timescale (*Harvey et al., 2013*). RA and PC input are thus integrated in such a way to culminate in temporally patterned cortical responses. One might expect that the shorter integration window of the PC filters combined

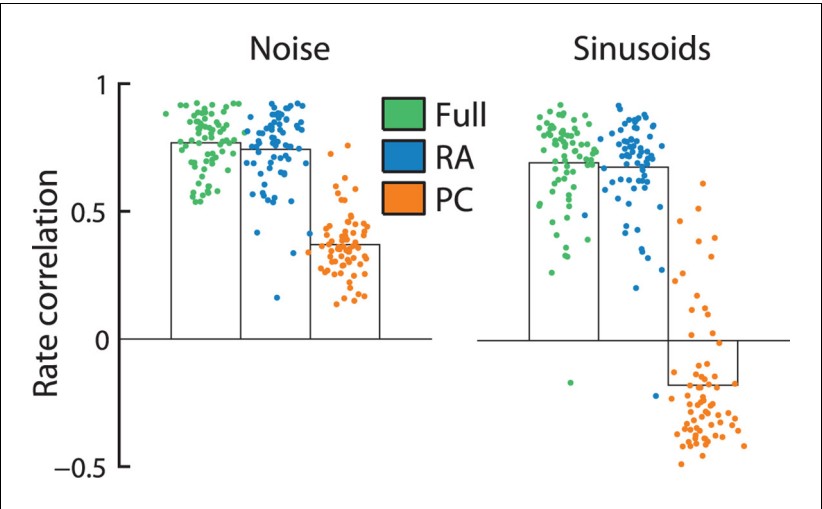

**Figure 3.** Rapidly adapting (RA) input determines cortical firing rates. Reconstruction accuracy of the full model compared to one that includes only RA or Pacinian (PC) input as measured by correlation between actual and predicted firing rates (measured over the duration of the trial, namely 1 s). Error bars denote the standard error of the mean across neurons. Firing rates elicited by both noise and sinusoidal stimuli are well predicted by the full model. While the performance of RA-only filters is almost as good as the full model, reconstruction accuracy drops dramatically if only PC input is included.

with their strong suppressive components would result in this sensory channel driving more precise spike timing than does its RA counterpart. Indeed, when removing the PC contribution from the model, predictions of spike timing in the cortex became less temporally precise (*Figure 4A*). We quantified this effect in two ways. First, we filtered the recorded and predicted time-varying firing rates to noise stimuli at different temporal resolutions and examined how much of the variance in the responses could be accounted for based on RA input or PC input alone (*Figure 4B*). We found that only PC input can account for high frequency fluctuations in the cortical responses of many neurons, while RA input is sufficient at coarser temporal resolutions for almost all cortical neurons (*Figure 4C*). In fact, the more precise the responses of a cortical neuron were, the more of its response variance was explained by PC input (*Figure 4—figure supplement 1*). Second, we tested how well the model could account for the well-documented patterning in cortical responses to sinusoidal vibrations: over a wide range of frequencies, cortical neurons produce entrained responses to sinusoidal stimulation of the skin (*Harvey et al., 2013*; *Salinas et al., 2000*; *Mountcastle et al., 1969*). That is, they produce one spike or burst of spikes within a very restricted phase of each stimulus cycle. We examined, then, whether we could predict the degree of entrainment of cortical neurons to sinusoidal stimuli from the RA and PC input. We found that the strength of phase-locking to sinusoids ranging in frequency from 50 to 300 Hz was well captured by the model (yielding a correlation between predicted and measured phase-locking of 0.78). Importantly, while RA input was dominant at the low frequencies, only the PC input could account for the observed phase-locking at 200 Hz and above (*Figure 4D*).

## RA and PC input is integrated optimally to process natural tactile scenes

Having established how input from different sensory channels is integrated in somatosensory cortex, we sought to understand *why* the two channels drive cortical neurons in different ways. We reasoned that the process of integration should reflect differences in the response properties of the input channels. Specifically, PC responses to skin vibrations and scanned textures have been consistently shown to be more temporally precise than their RA counterparts (*Weber et al., 2013*; *Mackevicius et al., 2012*). Furthermore, the two afferent classes differ in their frequency sensitivity profile: PC afferents are considerably more sensitive than RA afferents at frequencies above about 150 Hz (see *Figure 1A*). Finally, we surmised that the process of integration would also reflect the statistics of the stimuli experienced during interactions with objects, as is the case in other sensory modalities (*van Hateren, 1992*; *Field, 1987*; *Smith and Lewicki, 2006*).

We focused on natural textures, which are ideally suited to address this question. When running our finger across a textured surface, high-frequency oscillations are elicited in the skin (*Bensmaïa and Hollins, 2003*; *2005*; *Manfredi et al., 2014*), which in turn elicit highly temporally patterned and repeatable responses in both RA and PC afferents (*Weber et al., 2013*). In cortex, neurons respond to skin oscillations with precisely timed spikes that encode the skin deflections with millisecond precision (*Harvey et al., 2013*). In light of this, how should RA and PC responses to natural tactile stimuli be integrated to be maximally informative about texture-elicited skin oscillations at a fine temporal resolution? To address this question, we simulated RA and PC responses to different textures scanned across the skin and optimized cortical RA and PC filters for information transmission about skin oscillations.

We found that the filters optimized to convey information about texture closely matched the filters derived from measured S1 responses: RA filters were mostly excitatory and temporally broad; PC filters were narrower and comprised distinct suppressive components (*Figure 5A*). The similarities between the recorded and optimized filters also held up quantitatively. First, we found that most (71%) optimized filters comprised contribution from both RA and PC fibers, while the rest relied on RA input alone, similar to what we found in our cortical data set. Second, we found that the optimal filters typically weighted the RA input more strongly than the PC input, as did the filters derived from cortical responses (*Figure 5B*). Third, the optimal RA filter widths were wider than PC filter widths and largely overlapped the range of the actual filter widths (*Figure 5C*). Finally, we found that the excitation/suppression indices of the optimized filters approximated those derived from our cortical data set in that RA input was always net excitatory, while PC input spanned the range between highly suppressive, balanced, and excitatory (*Figure 5D*).

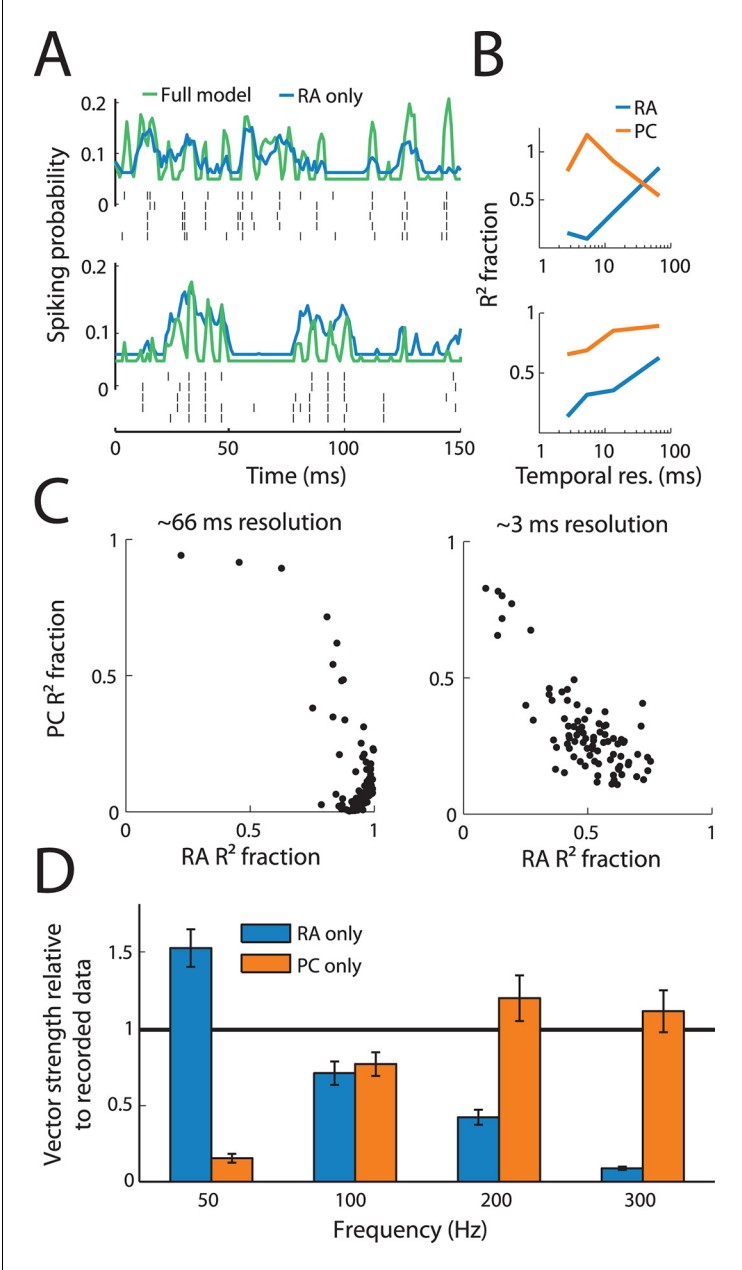

**Figure 4.** Precise spike timing of S1 neurons is driven by Pacinian (PC) input. (**A**) Responses to noise stimuli from two example neurons with particularly precise responses. Spiking probabilities derived from the full model (green traces) are much more temporally precise than are those derived from a rapidly adapting (RA)-only model (blue traces), reflecting the importance of PC input in shaping the response at fine timescales. Five repetitions of the recorded cortical spike trains are shown below the model traces in black. (**B**) Fraction of the explained variance ($R^2$) by either the RA (blue) or PC (orange) model relative to the full model across different temporal resolutions for the two neurons shown in A. PC input is important to explain responses on fine timescales. (**C**) Fraction of explained variance ($R^2$ fraction) relative to the full model by RA and PC input for all neurons at two different temporal resolutions. While RA input can explain most of the observed variance at coarse temporal resolutions (left panel), PC input is needed to explain the timing of cortical responses at fine temporal resolutions (right panel). See *Figure 4—figure supplement 1* for further analysis. (**D**) Vector strengths predicted by RA (blue) and PC (orange) models relative to their measured counterparts at different frequencies across all neurons. RA input accounts for entrainment to sinusoidal stimuli at low frequencies, while PC input is needed at higher frequencies.

The following figure supplement is available for figure 4:

**Figure supplement 1.** Fraction of variance explained as a function of timing precision of cortical neurons.

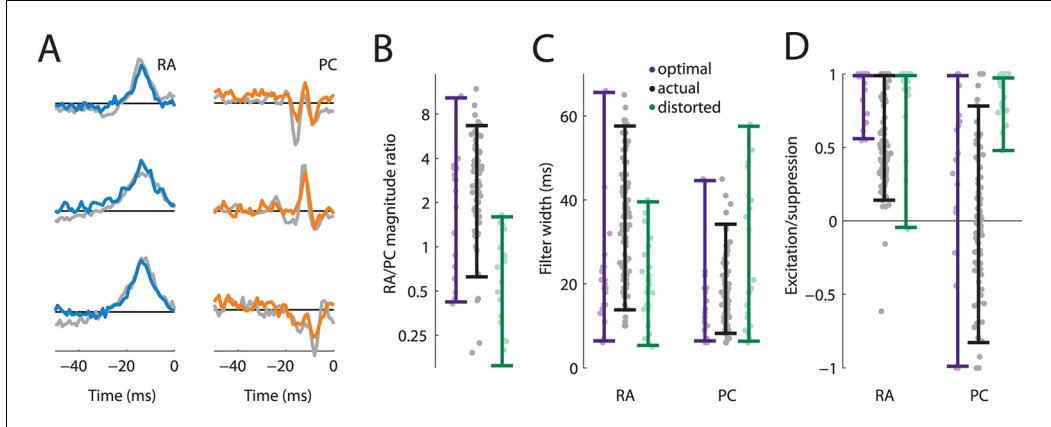

**Figure 5.** Rapidly adapting (RA) and Pacinian (PC) input is integrated optimally. (**A**) Three examples RA and PC filters, each optimized for a different natural texture (colored traces) and paired with a filter from a cortical neuron that most closely matches it (gray traces). (**B–D**) Comparison of filter statistics for actual (black) and optimized (lavender), and distorted (green) filters for the RA/PC magnitude ratio (B), filter width (C), and excitation/suppression index (D). For the optimized filters, bars denote the range covered by the filters optimized for different textures. For the actual filters, bars indicate the range between the 10th and the 90th percentile.

To test whether our fitted and optimized filters might be an artifact of our approach rather than reflect the integration properties of S1 neurons, we also optimized filters on a distorted data set. During natural interaction with surfaces, the amplitude of elicited vibrations decreases as their frequency increases, leading to roughly a power–law relationship (*Manfredi et al., 2014*; *Wiertlewski et al., 2011*; *Delhaye et al., 2012*). When we re-optimized the filters to stimuli with a distorted frequency composition, we found that the resulting filters did not match the ones observed in cortex (green bars in *Figure 5B–D*): The PC signal was weighted more heavily, was integrated over longer timescales, and was mostly excitatory, in contrast to what we observed in both the cortical filters and the ones optimized on our natural texture data set. We conclude that RA and PC integration in S1 is optimized to encode stimuli that are commonly encountered during interactions with objects.

## Discussion

We set out to examine how input from different mechanoreceptive afferent classes is integrated in the responses of individual cortical neurons. To this end, we simulated the responses of populations of RA and PC tactile afferents and examined how these are combined to drive S1 responses. This modeling effort yielded several important conclusions. First, the vast majority of cortical neurons receive (indirect) input from both RA and PC afferents. Second, RA and PC afferents drive S1 neurons in fundamentally different ways: RA input determines the rate of the response whereas the principal contribution of PC input is to sharpen the temporal precision of the response. Third, the integration of afferent input is consistent with a process that maximizes the information conveyed in the cortical responses by exploiting the different response properties of RA and PC afferents.

### Input from different sensory channels is integrated in the responses of individual S1 neurons

Our results unequivocally show that RA and PC input are integrated in the responses of individual S1 neurons. That is, the responses of S1 neurons cannot be explained from the responses of RA or PC neurons alone. While it has traditionally been assumed that the different tactile submodalities are segregated at least through primary somatosensory cortex (*Sur et al., 1981*; *Sur et al., 1984*; *Chen et al., 2001*; *Friedman et al., 2004*) (although cortical PC neurons have generally proven elusive, see below), recent evidence suggests that input from different classes of somatosensory afferents is indeed integrated between periphery and cortex (*Saal and Bensmaia, 2014*; *Pei et al.,*

*2009*; *Carter et al., 2014*; *Prescott et al., 2014*). In this study, stimuli fell in the range of frequencies that excites RA and PC fibers at the exclusion of SA1 fibers, which only respond to low-frequency stimulation. However, SA1 and RA input have been shown to be integrated in the responses of S1 neurons when these were studied with ramp-and-hold skin indentations (*Pei et al., 2009*) so the convergence of submodality-specific input onto individual S1 neurons seems to be a general phenomenon. However, it is still an open question whether the integration of input from different mechanoreceptors takes place only at the cortical level, or whether they appear already at the subcortical level in either the cuneate nucleus or the thalamus. Convergence of somatosensory submodalities has been observed in the brainstem of rats (*Sakurai et al., 2013*) and the same is likely true in primates.

## The effect of PC input on S1 responses

Our results reconcile some seemingly contradictory findings regarding the contribution of PC afferents to cortical responses. First, vibrotactile coding in general and the role of PC input specifically have been probed almost exclusively with high-frequency sinusoidal stimuli. These stimuli have been shown to evoke only a short transient response in S1, which then quickly decays back to baseline, a phenomenon that can be accounted for by the balance of excitation and suppression in the PC drive. That PC-elicited responses are short and weak might explain why PC-driven cortical neurons are thought to be extremely rare in S1 (*Lebedev and Nelson, 1996*; *Mountcastle et al., 1990*). Second, the main effect of PC input is to shape the timing of cortical spikes, which could account for the extremely precise timing of S1 responses at high frequencies (*Harvey et al., 2013*) that has been shown to contribute to tactile perception (*Zuo et al., 2015*). Third, the balanced drive of PC afferents explains why the firing rates of S1 neurons are not dependent on stimulus frequency even though PC rates are (*Harvey et al., 2013*). Finally, the PC input has been found to inhibit cortical responses (*Tommerdahl et al., 2005*; *Tommerdahl et al., 2010*), consistent with our finding that PC filters are often net suppressive.

## Efficient coding and the integration of sensory channels

Sensory cortex is tuned to the statistics of the natural environment, as has been demonstrated most notably in vision (*van Hateren, 1992*; *Field, 1987*) and audition (*Smith and Lewicki, 2006*), by invoking the efficient coding principle (*Barlow et al., 1961*). Generally, it has been assumed that the peripheral input stems from a single homogeneous population of neurons (e.g. retinal ganglion cells). However, this is often not the case: in touch, information is carried by multiple separate populations of neurons, each with different sensitivities and response properties. To encode information efficiently, then, the disparate response properties of these different channels need to be taken into account. While both RA and PC afferents respond to most high-frequency skin vibrations, such as those elicited during texture scanning, both afferent classes do so in different ways. First, RA afferents increase their firing rate with increasing stimulus amplitudes (*Arabzadeh et al., 2014*), and do so in a manner that is relatively independent of stimulus frequency, while PC responses saturate at low stimulus amplitudes and are highly frequency-dependent (*Muniak et al., 2007*). RA firing rates therefore convey a much more informative signal about the amplitude of tactile stimuli than do PC firing rates. That S1 firing rates depend mostly on RA input exploits the fact that these afferents convey unambiguous information about stimulus amplitude. Second, PC afferents exhibit much higher temporal precision in their spiking patterns than do RA afferents (*Mackevicius et al., 2012*), especially for high-frequency skin vibrations such as those elicited when we run our fingers across a texture (*Weber et al., 2013*), and this precise spike timing conveys information about tactile events and shapes perception (*Mackevicius et al., 2012*). That the timing of S1 responses is driven primarily by the PC input exploits the highly informative nature of the PC timing signal.

Separate sensory channels that convey complementary but overlapping information are commonplace in sensory systems and not limited to the sense of touch; the visual (*Field and Chichilnisky, 2007*), auditory (*Cant and Benson, 2003*), gustatory (*Zhang et al., 2003*), olfactory (*Uchida et al., 2014*), and vestibular (*Goldberg, 2000*; *Sadeghi et al., 2007*) systems all involve many types of afferents. Distributing sensory information across channels has distinct advantages such as parsing the behaviorally relevant range (*Dominy and Lucas, 2001*), keeping energy expenditure low (*Gjorgjieva et al., 2014*), and optimizing information transmission in the presence of noise

(*Kastner et al., 2015*). To the extent that these parallel input channels represent information in disparate ways (differing in response latency, adaptation properties, or spiking precision, among others), their integration should reflect and exploit such differences, a process that can be described using the type of model introduced here.

## Materials and methods

### Stimuli

The vibrotactile stimuli have been described previously, for both the peripheral (*Muniak et al., 2007*) and the cortical (*Harvey et al., 2013*) experiments. In short, tactile stimuli were delivered to the distal pads of the digits using a stainless steel probe diameter driven by a shaker motor. The shaker motor was calibrated before each experimental run such that stimuli were highly accurate and repeatable. We delivered three types of stimuli that varied in spectral complexity, including simple sinusoids, diharmonic stimuli (only used for the peripheral model fitting), and bandpass noise. The ranges of frequencies and amplitudes spanned the range experienced in everyday tactile experience. While the exact frequencies and amplitudes differed somewhat between the peripheral and cortical data sets, the two stimulus sets were largely overlapping. We were thus able to develop peripheral spiking models that simulated the peripheral responses to the stimuli used in the cortical data set (see below). Individual stimuli in this data set were 1 s long and were repeated five times (frozen noise), yielding a total of about 22 minutes (16 for noise and 6 for the sinusoids) of data from each neuron.

### Electrophysiology

#### Cortical experiments

Experimental procedures have been previously described (*Harvey et al., 2013*) and are only summarized here. All procedures were in accordance with the rules and regulations of the University of Chicago Animal Care and Use Committee. Extracellular recordings were made in the postcentral gyri in four hemispheres of two awake, behaving Rhesus macaques (*Macaca mulatta*). Recordings were obtained from neurons in areas 3b and 1 that met the following criteria: (1) action potentials were well isolated from the background noise, (2) the RF of the neuron was on the glabrous skin, and (3) the neuron was clearly driven by light cutaneous stimulation. In total, recordings were obtained from 49 area 3b and 69 area 1 neurons. Some of these data have been previously published (*Harvey et al., 2013*).

#### Peripheral experiments

Experimental procedures have been previously described (*Muniak et al., 2007*). All experimental protocols complied with the guidelines of the Johns Hopkins University Animal Care and Use Committee and the National Institutes of Health Guide for the Care and Use of Laboratory Animals. Single unit recordings were made from the ulnar and median nerves of two anesthetized Rhesus macaques using standard methods (*Talbot et al., 1968*). Recordings were obtained from 14 RA and 4 PC afferents. This data set has been previously used to develop peripheral spiking models (*Kim et al., 2010*; *Dong et al., 2013*).

### Peripheral model

We fit integrate-and-fire (IF) models to the measured responses of individual afferents. Our IF model is similar to an earlier one (*Kim et al., 2010*), with two main differences. First, we did not employ complex temporal filters with a large number of weights, but instead restricted the model input to six dimensions (positive and negative rectified stimulus positions, velocities and accelerations) plus a delay parameter, as this parameterization has been shown to be sufficient to reproduce afferent responses (*Dong et al., 2013*). Second, we included an additional parameter, namely a smoothing factor that controls the width of a Gaussian window that is convolved with the stimulus trace initially. Using this parameter allows us to model the fact that the PC afferents' sensitivity decreases for frequencies higher than ~300 Hz (*Muniak et al., 2007*; *Mountcastle et al., 1972*).

For model fitting, we used the *van Rossum* spike distance (*van Rossum, 2001*) between the recorded and model-predicted responses as a cost function and then optimized the model parameters using the *patternsearch* function in Matlab (The Mathworks, Inc., Natick, MA) using different

starting positions. We optimized the model on the noise stimuli first and then alternated between the sinusoidal and diharmonic stimulus sets while decreasing the cost function time constant until the fits did not improve further. In total, we fit single afferent models to 14 RA and 4 PC afferents. RA and PC population responses were generated by averaging over the responses of all RA and PC individual models, respectively. The models have been extensively validated, as detailed in previous studies (*Kim et al., 2010*; *Dong et al., 2013*), and captured both the strength and timing of peripheral afferents (*Figure 1—figure supplement 1*).

One potential concern is that the reconstructed RA and PC population activity might not accurately reflect the peripheral input on which a given cortical neuron's response relies. While individual peripheral responses are very stereotyped and therefore simple to model, individual afferents still differ in terms of sensitivity, and these differences might be reflected in the cortical responses. To address this issue, we generated different RA and PC populations, using different peripheral models, and also varied the size of the peripheral population used to drive the cortical responses. We found that the recovered filters were robust to these changes.

## Modeling the transformation between periphery and cortex

We modeled the transformation between the peripheral RA and PC population responses and single unit S1 responses using the LNP framework (*Schwartz et al., 2006*). LNP models have long been fruitfully used for modeling the responses of cortical neurons across several synapses, often even including the sensory receptors. The model describes the process of integration as follows: First, the peripheral RA and PC population responses are each convolved with a linear filter. Second, the resulting responses are summed and rectified to yield time-varying spiking probabilities. Third, spike trains are generated according to the spiking probability in each time bin. The full model (see also *Figure 1B*) can thus be expressed as follows:

$$p(s_t) = r(\boldsymbol{k}_{RA} \times \boldsymbol{s}_{RA,t} + \boldsymbol{k}_{PC} \times \boldsymbol{s}_{PC,t}) \tag{1}$$

where $p(s_t)$ denotes the probability of a cortical spike at time $t$, $\boldsymbol{k}_{RA}$ and $\boldsymbol{k}_{PC}$ are vectors containing the RA and PC filters, $\boldsymbol{s}_{RA,t}$ and $\boldsymbol{s}_{PC,t}$ are vectors containing the binned RA and PC population responses within a time window 100 ms prior to time $t$, and $r()$ denotes the rectifying nonlinearity.

To estimate the linear filters $\boldsymbol{k}_{RA}$ and $\boldsymbol{k}_{PC}$, we used reverse correlation of the cortical responses onto the RA and PC z-scored population responses covering a time window of 100 ms preceding the cortical response with a bin width of 1 ms. For this procedure, we exclusively used the data from the bandpass noise stimuli as this method can only recover unbiased filters if the RA and PC population responses are uncorrelated with each other over time. Even though our original vibrotactile stimuli exhibited little autocorrelation, the evoked RA and PC population responses were weakly correlated both across time and with each other, which could affect the filter estimation. We addressed this problem using a two-pronged approach. First, we verified that across our entire data set, temporal and population correlations were low (see below). Second, to minimize the impact of any remaining correlations, we calculated the autocorrelation matrix of the RA and PC population input and used its regularized inverse to correct the obtained filters (ridge regression):

$$\boldsymbol{k}_{RA,PC} = [\boldsymbol{S}^{\mathbf{T}}_{RA,PC,T} \times \boldsymbol{S}_{RA,PC,T} + a * \boldsymbol{I}]^{-1} \times \boldsymbol{S}^{\mathbf{T}}_{RA,PC,T} \times \boldsymbol{s}_{C,} \tag{2}$$

where $\boldsymbol{k}_{RA,PC,T}$ is a vector containing both the RA and PC filters, $\boldsymbol{S}_{RA,PC,T}$ is a matrix where each row corresponds to a time $t$ and contains the peripheral population responses $\boldsymbol{s}_{RA}$ and $\boldsymbol{s}_{PC}$ for the 100 ms preceding $t$, $a$ is the regularization parameter, $\boldsymbol{I}$ is the identity matrix, $\boldsymbol{s}_C$ is a binary vector containing the responses of a cortical neuron (0 for no spike, 1 for spike) for all times $t$, and $[]^{-1}$ denotes the pseudoinverse.

We varied the regularization parameter $a$ and also tried other correction methods, such as decomposing the autocorrelation matrix using singular-value-decomposition and only preserving strong components (*Sripati et al., 2006*); we obtained essentially the same results, and thus conclude that the approach is robust and the recovered linear filters accurate (see further analysis below).

The nonlinearity was a piecewise linear function, chosen over a smoother sigmoidal function, because the firing rates of S1 neurons increase steeply as soon as their respective response threshold is crossed (*Harvey et al., 2013*):

$$r(x)=b_1, \text{ if } x < 0, \; r(x) = b_1 + b_2 * x, \text{ if } x > 0 \text{ and } b_1 + b_2 * x < 1, \text{ and } r(x) = 1, \text{ otherwise.} \qquad (3)$$

To estimate the parameters $b_1$ and $b_2$, we used an optimization algorithm (*lsqcurvefit* in Matlab) to fit the time-varying spiking probabilities $p(s_t)$ such that they matched the observed spiking probabilities to the extent possible.

### Analysis of linear filters

To measure the magnitude of a filter, we summed its absolute value over time. To determine the temporal width of a filter, we first determined a noise threshold, set to three times the standard deviation of the filter values 80–100 ms before the cortical response; we then counted the total time that each filter's absolute value was above that threshold. Finally, to measure excitation/suppression indices, we divided the sum of the positive and negative components of a filter by its overall magnitude. When analyzing the impact of removing input from one fiber class, we refitted the filters and nonlinearity using only input from the remaining class.

### Testing the robustness and accuracy of filter estimation

Recovering linear filters accurately from data that exhibit correlations is notoriously difficult and error prone. In our case, one of the main concerns is whether cross-correlations between RA and PC responses might introduce biases in the recovered filters that would lead us to conclude that both populations contribute input, when in reality only one does. Additionally, the RA and PC responses are autocorrelated, which might bias our estimates of the filter width, or their net excitation or suppression, and thereby erroneously yield different filters for the two populations (see *Figure 2—figure supplement 1A* for the RA/PC covariance matrix). To test whether our method is prone to such biases, we simulated a population of 243 neurons with a variety of different linear filters that differed in width, magnitude, excitation/suppression, temporal offset, and received RA input, PC input, or both. We then used these filters to generate simulated spiking data from the noise vibrations. We set the firing rates to span the range encountered in the measured cortical responses. We then tested whether we could accurately recover the linear filters from the simulated responses and whether there were any biases in the measures we used to quantify these filters. First, we found that the recovered filters matched the filters used to generate the simulated data well (*Figure 2—figure supplement 1B,C*). Second, we verified that our procedure for determining whether neurons received RA input, PC input, or both, worked correctly in the vast majority of cases: 90% of the simulated neurons that received input from both RA and PC populations were identified as such. Importantly, only 3% of simulated neurons that received input from only one of the peripheral populations were misclassified as receiving input from both (*Figure 2—figure supplement 1D*). Finally, we also checked whether there might be biases when estimating filter metrics such as overall magnitude, filter width, and excitation/suppression indices in RA as compared to PC filters that might explain the observed differences. We found that these biases were generally very small and in no case could account for the effect sizes observed in the cortical data set (see comparison in *Figure 2—figure supplement 1E*). In summary, our procedure recovers most filter shapes accurately and the small biases that remain cannot account for the observed effect sizes.

### Timing analysis

To assess the contribution of RA and PC input at different timescales, we bandpass-filtered the measured and predicted cortical responses to four different frequency ranges, corresponding to different temporal resolutions (3, 5, 13, and 66 ms) and then calculated the correlation coefficients for each neuron. To assess spike timing for the sinusoidal data set, we calculated the vector strength (*Goldberg and Brown, 1969*) (a measure of how precisely a neuron's response is aligned with a sinusoidal stimulus) of the recorded and predicted cortical responses.

### Optimizing filters by maximizing information

We first simulated the responses of populations of RA and PC fibers to 55 natural textures (*Manfredi et al., 2014*). In brief, textured surfaces were scanned over the fingertip of human subjects at different scanning speeds, while elicited vibrations were recorded using a Laser Doppler vibrometer (Polytec OFV-3001 with OFV 311 sensor head, Polytec, Inc., Irvine, CA). For each texture,

we collected 240 (8 subjects × 3 speeds × 10 repetitions) 500-ms long vibration traces. The recorded traces were bandpass-filtered to between 50 and 800 Hz and then served as input to the peripheral RA and PC spiking models. To adjust for the distance-dependent decay of the vibrations and the size of the contact location, we added a global scaling factor for each the RA and PC traces which was set to ensure that the mean simulated RA and PC firing rates matched those recorded using the same set of textures (*Weber et al., 2013*). Indeed, across textures the recorded and simulated firing rates matched well ($r$ = 0.83 for RA and $r$ = 0.92 for PC responses), indicating that our simulated responses capture actual texture-elicited responses well. Some textures, mainly coarser one, elicit responses not only in RA and PC afferents, but also in SA1 afferents; we excluded textures with an average SA1 contribution >10 Hz from further analysis. Since we were interested in tactile environments that cause both RA and PC afferents to respond, we also excluded textures where the RA or PC average firing rate was below 20 Hz. In total, this left 25 textures, whose elicited vibrations where in the high-frequency range and excited RA and PC but not SA1 afferents.

Next, we optimized linear filters for each texture individually by maximizing the mutual information $I(S,V)$ between the responses of our model ($S$) and the texture-elicited vibration traces ($V$):

$$I(S,V) = \sum_S \sum_V p(s,v) \log\left[p(s,v)/p(s)p(v)\right] \qquad (4)$$

The probability of a spike in an S1 neuron, $p(s)$, can be calculated from the LNP model described above. To estimate $p(v)$, that is, the distribution of stimulus values, we divided the stimulus trace into 1 ms long bins and calculated the absolute value of the deflection amplitude at each time point. The resulting values were divided into 50 bins and $p(v)$ was set to the relative frequencies with which each of the 50 values appeared. We introduced a delay of 15 ms to mimic the response delays observed in cortical responses. For optimization, we calculated the gradient of the mutual information $I(S,V)$ analytically and then used a standard constrained optimization method (*fmincon* in Matlab). We restricted the average firing probability, $\Sigma_t\, p(s_t)/T$, of the optimized neurons to values observed in our recorded data set to prevent neurons from responding at unnaturally high rates (*Tkacik et al., 2010*). We found that filters optimized for different firing rates differed in a scale factor, but were otherwise identical. We also used a constant set of parameters for the static nonlinearity; again, we found that these did not affect the shape of the optimized filters but rather changed their scale.

Finally, we ran our procedure for filter optimization on a distorted data set. Specifically, we numerically computed the second derivative (accelerations) of the original texture traces described above and re-normalized them to the magnitude of the original traces before simulating peripheral responses to these stimuli and then optimizing the filters. The differentiated traces contain the same frequencies as the original traces, but the higher frequencies are weighted more strongly.

## Acknowledgements

We would like to thank Erika Dunn-Weiss for getting this work started and Benoit Delhaye and Justin Lieber for comments on a previous version of this manuscript.

## Additional information

### Funding

| Funder | Grant reference number | Author |
| --- | --- | --- |
| National Science Foundation | IOS 1150209 | Sliman J Bensmaia |

The funders had no role in study design, data collection and interpretation, or the decision to submit the work for publication.

### Author contributions

HPS, SJB, Conception and design, Analysis and interpretation of data, Drafting or revising the article; MAH, Conception and design, Acquisition of data, Drafting or revising the article

## Ethics

Animal experimentation: This study was performed in strict accordance with the recommendations in the Guide for the Care and Use of Laboratory Animals of the National Institutes of Health. All of the animals were handled according to protocol #72042, which was approved by the institutional animal care and use committee (IACUC) of the University of Chicago and the Johns Hopkins University.

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
