## [Decision Letter]

Thank you for submitting your work entitled "Rate and timing of cortical responses driven by separate sensory channels" for peer review at *eLife*. Your submission has been favorably evaluated by Eve Marder (Senior editor), a Reviewing editor (Andrew King), and three reviewers, one of whom, Stefano Panzeri, has agreed to reveal his identity.

The reviewers have discussed the reviews with one another and the Reviewing editor has drafted this decision to help you prepare a revised submission.

Summary:

In this study, data-based models were used to examine how information from different mechanoreceptive afferent classes is integrated in the responses of individual cortical neurons. The authors report that cortical activity reflects a combination of rapidly-adapting and Pacinian corpuscle inputs, with the former determining overall firing rate whereas the latter are represented by the precise timing of the activity. Their model results suggest that information about textures is represented in the primary somatosensory cortex by the statistically optimal integration of inputs from these two peripheral receptor channels.

Essential revisions:

Although the three reviewers agree that this is an interesting and important study and that the analysis is rigorous, they have raised the following issues.

1) Are the authors able to say where the inferred convergence of inputs from these two classes of mechanoreceptors takes place – in the cortex or at a subcortical level?

2) Given that model assessment largely revolves around predictive power and predictive power is impacted by reliability, the measurements of reliability mentioned in the Results (second paragraph) should probably be shown. Perhaps the authors could plot reliability vs accuracy. The authors should clarify how they measured reliability.

3) On a related note, factors unrelated to the stimuli, such as the stochasticity of spike generation, lead to a lack of reliability, which the authors comment on. The reader is told how well the model predicts the total variance (about half), but the model is probably performing better than this since some of the variance is due to the lack of reliability in the responses. The authors should estimate how much of the predictable variance (rather than the total) the model accounts for (e.g., Sahani & Linden 2003 NIPS; Ramirez et al. 2014 Nat Neurosci).

4) The authors report (Figure 2) that most cortical neurons receive "significant contributions" from RA and PC populations. However, if the RA and PC population responses are correlated, a model-fitting algorithm might generate positive weights from a non-connected PC population even though cortical neurons receive input exclusively from RA afferents. It is particularly hard to exclude the possibility that a small (but statistically significant) input might be an artefact. The authors consider this issue in the Materials and methods. They state that the correlations were "low" – but how low? What is the correlation coefficient between the RA and PC population responses as a function of time resolution? Also, the authors use a whitening procedure to attempt to decorrelate (is equation 2 correct? Is the regularizing term not added?), but these procedures are delicate and it is not clear that the correlation issue has been fully dealt with. Further work is needed to address this. It may be useful to use a simulation approach, i.e. simulate cortical neurons with realistic filters that range from single channel to mixed, analyze them with their methods, and test whether their single/mixed nature is accurately recovered. Furthermore, the results (Figure 2) are based on a statistical test (no details are provided for how this was done or what the critical P value was). Hence, a neuron that counts as convergent might have a strong input from RA together plus a tiny PC input that passes the statistical test, but is functionally irrelevant. Suppose that you define convergent input as where at least 10% of the total input strength comes from each channel, do you still see the same prominent convergence shown in Figure 2?

5) The authors claim that cortical (time-averaged) firing rates are due to RA, not PC input. They state that their stimuli "drove RA and PC to the exclusion of SA1". On this basis, they consider only RA and PC responses in their study and ignore SA. However, Figure 1–Figure 2 of the supporting reference (Muniak et al., 2007) do show SA responses to both sinusoidal and noise stimuli. Given these data, it does not seem appropriate to ignore the SA afferents. Do the authors have peripheral recordings of SA1 responses? If so, they could apply their model-fitting approach. If they are correct, the weights from the SA population should turn out to be zero.

6) For the analyses shown in Figure 3–Figure 4 where full and RA/PC only models are compared, how were the RA/PC only results obtained? Were the cortical models retrained on only RA/PC input or were the RA/PC contributions in the full model simply set to zero? Given the concern about correlations, the authors should check whether they get the same results using the retraining method if that is not already what they did.

7) The conclusion that the precise timing of S1 neurons is mainly driven by PC input depends on how good the models for the RA/PC afferents are. The authors show a single neuron example (Figure 4) where removing PC input degrades timing precision, but in Figure 4 (right – the 3 ms resolution case) it is only for a minority of points that PC explains more variance than RA. In fact, for most of the cortical neurons, RA explains more variance. If correct, this seems to undermine the conclusion that PCs drive cortical spike timing. Figure 1—figure supplement 1 shows that the RA afferent model performs poorly at 3 ms resolution. In this case, how is it possible (in Figure 4 right) that they explain so much variance? (It would be useful to define the R squared measure precisely and exactly what "variance" is meant).

8) The authors report that "filters optimized to convey information about texture closely matched the filters derived from measured S1 responses". This is not currently fully convincing. It seems from the Materials and methods that they find the RA and PC filters that maximize mutual information between the stimulus (measured in a 1 ms bin) and the model's response in that bin. They repeat this for 25 different textures. Then they look through these 25 results and pick out the one that looks most like the filters fitted from spike data (this important part of the process is only mentioned in the legend to Figure 5 – it should be in the Materials and methods). There are several concerns with this approach.

A) In classic studies of this kind (e.g. Olshausen & Field, Nature, 2006; Attick, Network, 1992), the optimization is carried out by averaging across a bunch of natural stimuli, not on individual stimuli. It is possible that cherry-picking from 25 separate optimizations could result in spurious matches. It would be much more convincing if the optimization was carried out across the whole set of 25 textures.

B) Even using the texture-picking method, "close" matches are only shown for 3 examples in Figure 5. Figure 5 show similarities between the distribution of width and other parameters between real and optimized filters, but do not show how close these values are on a neuron by neuron basis (or even the means, medians or shapes of these distributions). If there really is systematic, close matching (which would be impressive), there should be a high correlation coefficient between e.g. actual and optimized filter width, etc. across neurons. Some test or comparison with other models is needed to justify the conclusion that filters for the recorded cells are/aren't optimal. Perhaps the authors could distort the textures or shuffle them somehow to see whether they can create "suboptimal" filters.

C) For the mutual information calculation, is it correct that stimulus and response are measured in the same simultaneous time bin? Should response latency not be taken into account?

9) For fitting the afferent models, how did the authors avoid over-fitting? Was cross-validation used when assessing model accuracy in Figure 1—Figure 1? Why the unusual approach of sequentially fitting to individual stimulus types – why not fit on all 3 stimulus sets at once?

10) It is stated in the first paragraph of the Results section that the stimuli for the cortical experiments were "analogous but not identical" to the peripheral ones. This is potentially a problem, since statistically fitted, nonlinear models cannot be relied on to extrapolate outside their training set. At present, there are insufficient methodological details to understand how different the stimuli really were. Please detail, in the Materials and methods, exactly what the stimuli were in the two cases. Related to this, it is stated in the Materials and methods that the peripheral recordings were done in anesthetized animals. What about the cortical ones?

11) Materials and methods, subsection “Electrophysiology”. The paper includes only 4 PC afferents. The authors need to justify why this small sample size is sufficient or increase the number. Similarly, it is unclear whether multiple peripheral models can be generated using only 4 recorded afferents (subsection “Peripheral model”).

---

## [Author Response]

*Essential revisions:*

*Although the three reviewers agree that this is an interesting and important study and that the analysis is rigorous, they have raised the following issues. 1) Are the authors able to say where the inferred convergence of inputs from these two classes of mechanoreceptors takes place – in the cortex or at a subcortical level?*

The model allows us to determine that the integration of signals from different afferent classes is taking place somewhere between the periphery and cortex, but not whether it is taking place in the cuneate nucleus, the thalamus, or in S1 neurons. Not enough is known about the integration properties of the two subcortical structures to make much of an educated guess either. We have now clarified this by adding the following passage to the Discussion section:

"However, it is still an open question whether the integration of input from different mechanoreceptors takes place only at the cortical level, or whether they appear already at the subcortical level in either the cuneate nucleus or the thalamus. Convergence of somatosensory submodalities has been observed in the brainstem of rats (Sakurai et al., 2013) and the same is likely true in primates."

*2) Given that model assessment largely revolves around predictive power and predictive power is impacted by reliability, the measurements of reliability mentioned in the Results (second paragraph) should probably be shown. Perhaps the authors could plot reliability vs accuracy. The authors should clarify how they measured reliability.*

We now measure reliability in the form of predictable variance (see our response to point 3 for more details) rather than using our own measure and we have revamped the section dealing with model assessment accordingly. A new plot (Figure 1—figure supplement 2) has been added to show predictable variance as a function of temporal resolution.

*3) On a related note, factors unrelated to the stimuli, such as the stochasticity of spike generation, lead to a lack of reliability, which the authors comment on. The reader is told how well the model predicts the total variance (about half), but the model is probably performing better than this since some of the variance is due to the lack of reliability in the responses. The authors should estimate how much of the predictable variance (rather than the total) the model accounts for (e.g. Sahani & Linden 2003 NIPS; Ramirez et al. 2014 Nat Neurosci).*

We have switched to using predictable variance for model assessment, which has the added benefit that it makes model assessment more comparable across different temporal resolutions. A new figure (Figure 1—figure supplement 2) showing predictable variance and the fraction of the predictable variance across different temporal resolutions.

The new section on model assessment now reads as follows:

"We found that our model captures the cortical responses well on both coarse and fine timescales as assessed by calculating the correlation coefficients between the predicted and the recorded responses. […] In other words, the model was able to predict neuronal responses with considerable accuracy given their inherent variability."

*4) The authors report (Figure 2) that most cortical neurons receive "significant contributions" from RA and PC populations. However, if the RA and PC population responses are correlated, a model-fitting algorithm might generate positive weights from a non-connected PC population even though cortical neurons receive input exclusively from RA afferents. It is particularly hard to exclude the possibility that a small (but statistically significant) input might be an artefact. The authors consider this issue in the Materials and methods. They state that the correlations were "low" – but how low? What is the correlation coefficient between the RA and PC population responses as a function of time resolution? Also, the authors use a whitening procedure to attempt to decorrelate (is equation 2 correct? Is the regularizing term not added?), but these procedures are delicate and it is not clear that the correlation issue has been fully dealt with. Further work is needed to address this. It may be useful to use a simulation approach, i.e. simulate cortical neurons with realistic filters that range from single channel to mixed, analyze them with their methods, and test whether their single/mixed nature is accurately recovered. Furthermore, the results (Figure 2) are based on a statistical test (no details are provided for how this was done or what the critical P value was). Hence, a neuron that counts as convergent might have a strong input from RA together plus a tiny PC input that passes the statistical test, but is functionally irrelevant. Suppose that you define convergent input as where at least 10% of the total input strength comes from each channel, do you still see the same prominent convergence shown in Figure 2?*

We followed the excellent suggestion to test our procedure for estimating the filters using simulated neuronal data. We have added a new section in the Materials and methods entitled "Testing the robustness and accuracy of filter estimation" that describes the new tests in detail, along with a new figure (Figure 2—figure supplement 1). In short, this analysis shows that we are able to recover linear filters accurately and that our metrics are only minimally affected by artifacts.

Our criterion for whether a neuron has input from only RAs, only PCs, or both was based on the whether the filters deviated by more than three standard deviations from the noise distribution (measured 100 ms away from the spike) for more than 5 ms. The reviewers are correct that small but significant deviations from the noise distribution (such as introduced by artifacts driven by the RA/PC cross-correlations) might lead to spurious results; indeed, our toy neuron analysis exhibited a number of conditions where this could happen. We thus added another criterion (the one suggested by the reviewers) which states that a neuron can only be classified as receiving input from both classes when the ratio of their filter strengths is not greater than 1:10. In our toy neuron analysis, this additional criterion led to almost no errors in input classification (see new Figure 2—figure supplement 1). However, this criterion was already met by the neurons in our data set (cf. old Figure 2) and so the results are unaffected.

Thank you for spotting the typo in equation (2); the regularization term should indeed be added.

The new section in the Materials and methods is reproduced here:

"Testing the robustness and accuracy of filter estimation

Recovering linear filters accurately from data that exhibit correlations is notoriously difficult and error prone. […] In summary, our procedure recovers most filter shapes accurately and the small biases that remain cannot account for the observed effect sizes."

*5) The authors claim that cortical (time-averaged) firing rates are due to RA, not PC input. They state that their stimuli "drove RA and PC to the exclusion of SA1". On this basis, they consider only RA and PC responses in their study and ignore SA. However, Figure 1–Figure 2 of the supporting reference (Muniak et al., 2007) do show SA responses to both sinusoidal and noise stimuli. Given these data, it does not seem appropriate to ignore the SA afferents. Do the authors have peripheral recordings of SA1 responses? If so, they could apply their model-fitting approach. If they are correct, the weights from the SA population should turn out to be zero.*

Our peripheral data set spanned more frequencies than the cortical data set. Specifically, the lowest frequency used in the cortical data set (and therefore in simulations and analyses of the current paper) was 50 Hz. The figures that the reviewers are referring to both show SA1 responses below 50 Hz and are therefore not applicable for the current study. A better figure to look at in reference (Muniak et al., 2007) would be Figure 4 which shows some (weak) SA1 activation even for frequencies above 50 Hz given rather high stimulus amplitudes (with thresholds generally around just below 100 microns). A valid concern is that these SA1 responses (which are currently not accounted for in the model) could bias the estimation of the RA and PC filters. We therefore ran our filter estimation again, this time restricting our data set to noise traces with an RMS amplitude of <50 microns; this conservative threshold all but excludes any possible contribution of SA1 afferents. While the resulting filters were slightly noisier (because fewer data were used to estimate them), all metrics such as the fraction of neurons with convergent input, the filter magnitude ratios, the filter widths, and the excitation/suppression indices almost exactly matched the ones calculated from the full data set. We thus conclude that any SA1 contribution does not affect our results. While we would have liked to also derive SA1 filters using our model, the fact they fire so sparsely to these stimuli means that their contribution to the S1 firing rates is minimal.

*6) For the analyses shown in Figure 3–Figure 4 where full and RA/PC only models are compared, how were the RA/PC only results obtained? Were the cortical models retrained on only RA/PC input or were the RA/PC contributions in the full model simply set to zero? Given the concern about correlations, the authors should check whether they get the same results using the retraining method if that is not already what they did.*

We were already using refitted filters for the RA/PC-only models. Compared to the full models, the filters were essentially the same. The only consistent difference was in the fitted nonlinearity, which generally needs to be steeper when only RA or PC input is used in order to achieve the same firing rates as the full model. We have now clarified this in the Materials and methods section.

*7) The conclusion that the precise timing of S1 neurons is mainly driven by PC input depends on how good the models for the RA/PC afferents are. The authors show a single neuron example (Figure 4) where removing PC input degrades timing precision, but in Figure 4 (right – the 3 ms resolution case) it is only for a minority of points that PC explains more variance than RA. In fact, for most of the cortical neurons, RA explains more variance. If correct, this seems to undermine the conclusion that PCs drive cortical spike timing. Figure 1—figure supplement 1 shows that the RA afferent model performs poorly at 3 ms resolution. In this case, how is it possible (in Figure 4 right) that they explain so much variance? (It would be useful to define the R squared measure precisely and exactly what "variance" is meant).*

First, the main goal of our analysis was to show that PC input is needed to explain the precise timing of cortical neurons. As such, the important comparison is not whether PC input explains the cortical responses better than RA input, but that RA input alone is not enough to explain the cortical responses at fine temporal resolutions. Second, not all cortical neurons exhibit precise spike timing (see Harvey et al., PloS Biol, 2013). Indeed, cortical neurons that exhibit more precise spike timing also depend more on PC input, as we now show in the new Figure 4—figure supplement 1.

We have added a sentence in the Results section to clarify this point:

"We found that only PC input can account for high frequency fluctuations in the cortical responses of many neurons, while RA input is sufficient at coarser temporal resolutions for almost all cortical neurons (Figure 4). In fact, the more precise the responses of a cortical neuron were, the more of its response variance was explained by PC input (Figure 4—figure supplement 1)."

*8) The authors report that "filters optimized to convey information about texture closely matched the filters derived from measured S1 responses". This is not currently fully convincing. It seems from the Materials and methods that they find the RA and PC filters that maximize mutual information between the stimulus (measured in a 1 ms bin) and the model's response in that bin. They repeat this for 25 different textures. Then they look through these 25 results and pick out the one that looks most like the filters fitted from spike data (this important part of the process is only mentioned in the legend to Figure 5 – it should be in the Materials and methods). There are several concerns with this approach.*

*A) In classic studies of this kind (e.g. Olshausen & Field, Nature, 2006; Attick, Network, 1992), the optimization is carried out by averaging across a bunch of natural stimuli, not on individual stimuli. It is possible that cherry-picking from 25 separate optimizations could result in spurious matches. It would be much more convincing if the optimization was carried out across the whole set of 25 textures.*

Olshausen & Field and similar approaches ask how information can be efficiently encoded in a population of neurons, and they optimize for both information transmission and sparseness. We, however, are concerned with the response properties of individual neurons in this paper, and ask whether we can explain the way single neurons integrate information from different inputs by a simple principle of information maximization. The reviewers are right to wonder whether filters similar to the cortical ones might not be trivially obtained by this method, and we have now added a new analysis that we think addresses this question (see below). We have also clarified the rationale for our approach in the Results section:

"We focused on natural textures, which are ideally suited to address this question. […] To address this question, we simulated RA and PC responses to different textures scanned across the skin and optimized cortical RA and PC filters for information transmission about skin oscillations."

*B) Even using the texture-picking method, "close" matches are only shown for 3 examples in Figure 5. Figure 5 show similarities between the distribution of width and other parameters between real and optimized filters, but do not show how close these values are on a neuron by neuron basis (or even the means, medians or shapes of these distributions). If there really is systematic, close matching (which would be impressive), there should be a high correlation coefficient between e.g. actual and optimized filter width, etc. across neurons. Some test or comparison with other models is needed to justify the conclusion that filters for the recorded cells are/aren't optimal. Perhaps the authors could distort the textures or shuffle them somehow to see whether they can create "suboptimal" filters.*

We have followed the reviewers' suggestions and now derived optimal filters also for a distorted data set, namely one where the usually observed power law between frequency and amplitude of skin oscillations has been broken. We find that the optimal filters derived for these non-natural scenes do not match the cortical ones as well as those derived from natural scenes. Furthermore, we do not want to make any claims that the optimal filters match the cortical ones on a neuron-by-neuron basis, but simply that similar filter properties are observed in the optimal and the actual filters, when we former ones are optimized on a data set containing natural scenes. We have added the following text in the Results section:

"To test whether our fitted and optimized filters might be an artifact of our approach rather than reflect the integration properties of S1 neurons, we also optimized filters on a distorted data set. […] The PC signal was weighed more heavily, was integrated over longer timescales, and was mostly excitatory, in contrast to what we observed in both the cortical filters and the ones optimized on our natural texture data set."

Additionally, we have added the following notes in the Materials and methods section:

"Finally, we ran our procedure for filter optimization on a distorted data set. […] The resulting traces are thus similar but differ in one important aspect (namely frequency composition) from the natural traces."

*C) For the mutual information calculation, is it correct that stimulus and response are measured in the same simultaneous time bin? Should response latency not be taken into account?*

Our analysis assumes a fixed response latency of 15 ms, comparable to what is observed in the cortical responses. This is now clarified in the Materials and methods.

*9) For fitting the afferent models, how did the authors avoid over-fitting? Was cross-validation used when assessing model accuracy in Figure 1—Figure 1? Why the unusual approach of sequentially fitting to individual stimulus types – why not fit on all 3 stimulus sets at once?*

The afferent models comprise few parameters so overfitting is not a serious concern. We have validated the approach extensively in previous published work, in which we systematically cross-validated the fits. Furthermore, the responses of peripheral afferents are very stereotyped and have been extensively described in the literature, so it is easy to spot erroneous response patterns.

Our approach of sequentially alternating between different data sets was implemented mainly to speed up the convergence of the parameters. Similar approaches, where parameters are fitted on alternating subsets of the data, are used in the machine learning community (for example in stochastic gradient descent algorithms).

*10) It is stated in the first paragraph of the Results section that the stimuli for the cortical experiments were "analogous but not identical" to the peripheral ones. This is potentially a problem, since statistically fitted, nonlinear models cannot be relied on to extrapolate outside their training set. At present, there are insufficient methodological details to understand how different the stimuli really were. Please detail, in the Materials and methods, exactly what the stimuli were in the two cases. Related to this, it is stated in the Materials and methods that the peripheral recordings were done in anesthetized animals. What about the cortical ones?*

Both peripheral and cortical stimulus parameters have been described in previously published papers that are referenced in the Materials and methods section. In both cases, simple sinusoids as well as bandpass noise traces were presented. While the exact frequencies and amplitudes differed somewhat between the peripheral and cortical data sets, there was broad overlap between the two sets. Thus, the peripheral models were only used to simulate responses to stimuli that were reasonably close to those encountered during their training. The models were further validated by assessing how well they could replicate well-known response properties of peripheral afferents, such as phase-locking and entrainment plateaus, some examples of which are shown in Figure 1—figure supplement 1. As mentioned above, we documented a much more systematic validation of these models in previous papers. The cortical recordings were from awake animals. This is now clarified in the text.

We have added the following sentence to the 'Stimuli' section in the Materials and methods:

"While the exact frequencies and amplitudes differed somewhat between the peripheral and cortical data sets, the two stimulus sets were largely overlapping. […] Individual stimuli in this data set were 1 s long and were repeated 5 times (frozen noise), yielding a total of about 22 minutes (16 for noise and 6 for the sinusoids) of data from each neuron."

*11) Materials and methods, subsection “Electrophysiology”. The paper includes only 4 PC afferents. The authors need to justify why this small sample size is sufficient or increase the number. Similarly, it is unclear whether multiple peripheral models can be generated using only 4 recorded afferents (subsection “Peripheral model”*).

We believe that 4 PC afferent models are sufficient for the following reason. We have more models available for RA afferents and can therefore test how population size affects our prediction accuracy. We find that using only one or two models leads to decreased performance (accounting for 40-50% of explainable variance at a resolution of 100 ms), but that accuracy levels off when at least 4 models are used (57%) and does not increase further with more afferents added (58% using all 14 RA models).